# The Sensitivity Analysis of the Drainage Unsteady Equations against the Depth of Drain Placement and Rainfall Time at the Shallow Water-Bearing Layers: A Case Study of Markazi Province, Iran

**Behzad Moshayedi** [1] , **Mohsen Najarchi** [1,*], **Mohammad Mahdi Najafizadeh** [2] **and Shahab Khaghani** [3]

1   Department of Civil Engineering, Islamic Azad University of Arak Branch, Arak 38135-567, Iran
2   Department of Mechanical Engineering, Faculty of Engineering, Islamic Azad University of Arak Branch, Arak 38135-567, Iran
3   Department of Plant Breeding, Islamic Azad University of Arak Branch, Arak 38121-411, Iran
*   Correspondence: m-najarchi@iau-arak.ac.ir; Tel.: +98-9188621836

**Abstract:** This research investigated various drainage parameters for unsteady conditions, including depth of installation, reflection coefficient, and depth of water table. For this purpose, Bouwer & Van Schilfgarrd, Dumm, Glover, Hemmad, and Bouwer equations were used. For the distance of computed drainage compared with measured data in central Iran, the results showed that the Bouwer & Van Schilfgaarde equation is better than others. Additionally, the installed depth was obtained 130 cm below the exiting underground, and this depth was applicable more than other depths; 1, 3, and 5-day precipitation were used to determine water table changes. The results illustrated that a 5-day duration had a better effect, which appeared in the existing condition drainage area. The reflection coefficient for the superior equation was also obtained as 0.65, which was very close to the measured data in the area. Mean Absolute Error (MAE), Root Mean Squared Error (RMSE), and Standard deviation ($\sigma$) were used to evaluate the results. MAE, RMSE, and $\sigma$ were computed as 1.78, 2.02, and 0.02, for the superior equation respectively, and the appropriate distance between the two drains was determined as 51.26 m. The obtained results have a close agreement with other researchers in this regard.

**Keywords:** hydraulic conductivity; unsteady conditions; reaction modulus; correlation coefficient; drain spacing

## 1. Introduction

With the increasing population and accelerated urbanization, demands for water are rising for different sectors around the world [1]. In the last two decades, water has clearly moved from the purview of experts and engineers to a wider forum where stakeholders articulate different claims, values, and interests around water management issues [2]. Water is one of the most scarcely available natural resources on earth. Its effective utilization is a major concern felt worldwide. The water crisis is a situation felt in every sector using it (domestic, power, industry, and irrigation). The exploitation of water resources in an unplanned manner has resulted in the deterioration of environmental and social conditions and has resulted in an alarming situation [3]. Using reclaimed water for agricultural irrigation is one of the most important measures for alleviating the global water crisis [4]. The increasing scarcity of water in dry areas is now a well-recognized problem. According to the World Commission on Environment and Development, approximately 80 countries with 40% of the world population already suffer from serious water shortages [5].

Population growth and the increasing desire for urbanization and consumerism have increased the human need for agricultural products [6]. Thus, attention to agriculture and water resource management is increasing worldwide [7]. A lack of accessible drinking

water resources is a major threat to human survival, the sustainability of natural ecosystems and sustainable urban development, and the food, health, and macroeconomic security of any society is highly vulnerable to water scarcity. Currently, more than 90% of controlled water in developing countries, including Iran, is used to irrigate agricultural land, and the rest is used for drinking and industry in rural and urban areas [8]. Research has shown that not paying attention to proper drainage management will lead to salinity and the destruction of agricultural lands [9]. Land drainage will not be possible without considering irrigation methods and the optimal use of water resources [10]. Any irrigation project will be successful if it is implemented in the area at the same time as considering the appropriate drainage plan [11]. In other words, irrigation and drainage are interdependent [12]. Otherwise, many agricultural lands will certainly be destroyed. For subsurface drains to work well, they must be well designed and implemented using appropriate numerical models to meet the water needs of the area [13].

A disproportionate distribution of drinking water is very worrisome. Urban population growth and an increase in the need for water resources have led water resources to be more threatened in terms of quantity and quality. Among other factors exacerbating the water crisis, urban consumerism, increased desire to consume organic foods, rapid population growth, demand for social welfare, and an increase in the level of recreational and welfare needs of society can be considered. Drainage systems have historically been used to remove excess water from the land surface. In addition, in the last two decades, managerial and engineering attitudes have been considered in this field so that drainage waters can be used to deal with the water crisis in other lands. Recently, there have been growing concerns about the optimal use of drainage models in communities following the water crisis, and environmental protection has been identified as the largest drainage challenge in the world. The drainage of agricultural lands is known as one of the most important factors of the production rate and increasing yields of horticultural crops and crops in areas with dry and wet climates. However, this issue has been neglected in many areas. Innovative methods for the integrated design and management of groundwater irrigation and drainage systems have tremendous potential for improving efficiency, helping to meet food needs in facing an increase in population growth and a decrease in global freshwater resources.

According to the above explanations, the researchers try to identify the available water resources in each part of the land, to use water resources, even though on a small scale, and to be effective in reducing the water crisis and optimal use of water resources by implementing optimal drainage models and creating the right bed for transferring excess water to reservoirs. Land drainage can be effective in preventing water retention in agricultural lands, which leads to reduced agricultural production efficiency, or in preventing the formation of seasonal runoff, which is generally caused by heavy seasonal rainfall and damages buildings and facilities (development projects). A unique feature of the study area in this paper is the presence of a rock bed on the surface of the land, which has caused the depth of the water-impermeable layer to be drastically reduced, and due to the existence of a rock bed and high drilling costs, it is not possible to design and place underground drains at the same depth. Additionally, due to the cold and dry climate of the studied region, which is formed due to the mountainous region in the northern part of the Senejan Plain, surface water resources by the formation of seasonal runoff, as well as a groundwater flow caused by groundwater aquifers, have led to the destruction of buildings and facilities (development projects) in the northern part and waterlogging on the land surface of agricultural lands in the southern part, which has reduced the production of agricultural products and consequently has created some social challenges in this sector in the past decade.

In this article, the researcher has investigated the distance between the drains by using numerical drainage models for conditions where the drains are located at different depths. The research area includes 2 northern and southern parts. The northern part includes residential areas and the southern part includes agricultural lands. The goal of

the researcher in conducting this research is to be able to reduce the cost of the project by reducing the depth of digging and installing underground drainage pipes. This reduction in the amount of drilling will also increase the efficiency of project implementation. With the correct implementation of drainage, the surface and underground waters of the northern part of the land can be transferred to the southern parts and agricultural lands in the shortest possible time. This management in the correct transfer of surplus water can lead to the guarantee of sustainable urban and agricultural development in the study area.

## 2. Studied Region

The region in this study is a part of Markazi province in terms of political divisions, which is located in the central part of Arak city in Iran. This urban watershed, with an area of more than 70 hectares, is located between the geographical coordinates of latitude: 34.08652 and longitude: 49.688842 [14], next to the watershed of the Gharah Kahriz region, and the main discharge of water leads to the Meighan wetland. Additionally, to obtain numerical and statistical information about the meteorology of the region, the synoptic station of Arak city, which is located at an altitude of 1708 m above sea level, was used. This station is located at 46–49 degrees' east longitude and 34–36 degrees north latitude. This station is the closest meteorological station to the studied region, and meteorological data from this station can be used. The aerial image of Figure 1 shows the location of the region. According to the figure, the northern part consists of lands with residential, commercial, and service uses and is adjacent to the mountain range called Ghassbe, and the southern part, known as the Senejan Plain, is covered with agricultural lands with a variety of crops.

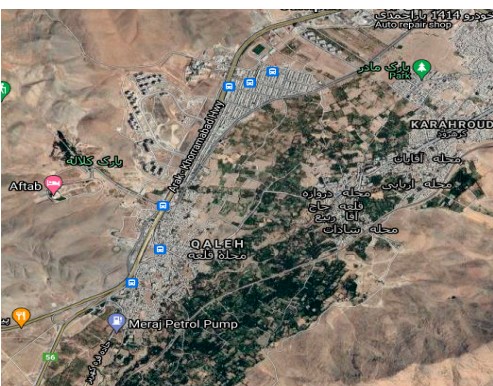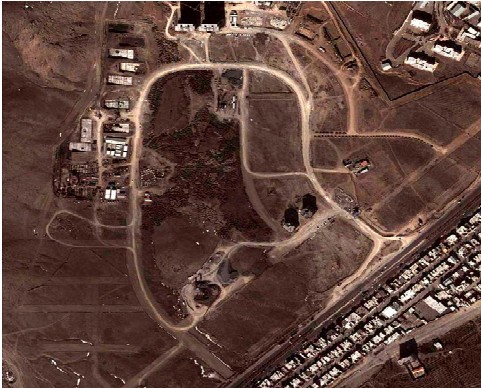

**Figure 1.** Aerial map of the study area in the Sanjan Plain. Right image of northern lands with residential use. Left image with agricultural use in the southern part.

The lands of the Senejan Plain are located at the westernmost point of Arak city. In these lands, due to the existence of surface water resources in the rainy season as well as groundwater resources, surface runoff is formed. By creating a bed for optimal and correct management, environmental challenges and multiple damages to the lands should be prevented. Among the regional challenges, the lack of optimal management regarding the control of damages due to runoff during the last years can be mentioned. The lands in the northern part consist of residential and commercial buildings and facilities. During the 10 years of exploitation of the northern lands, we have witnessed many financial and social damages caused by 1- or 2-day seasonal rainfall. Two-day rains in the rainy seasons have led to uncontrolled movement of surface waters, runoff formation, improper transfer to southern lands, water retention in agricultural lands, and widespread destruction of agricultural lands.

Figure 2 shows the water retention in facilities and building beds that flooding from surface water during 2-day rainfall has been able to flood more than 10 m from the height of a building under construction. Figure 3 shows the water retention in agricultural lands

in the southern part due to 3-day rainfall in 2019, which is the result of the transfer of excess water from the northern lands and the uncontrolled transfer to the southern lands.

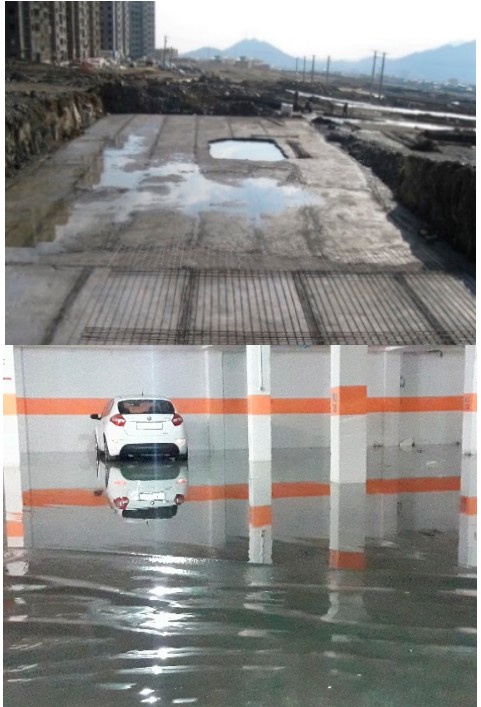
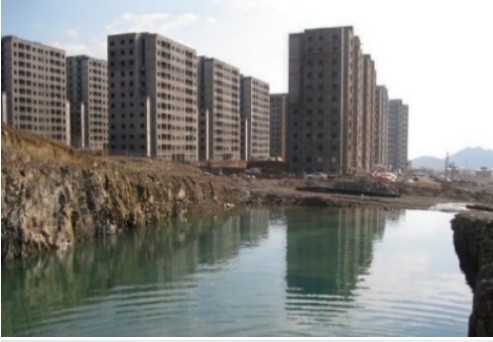
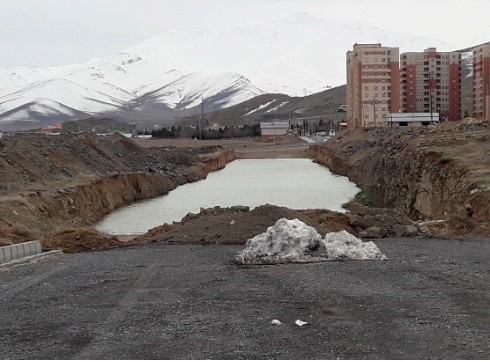

**Figure 2.** Water accumulation in the northern part of the study area with residential use during 2 days of rainfall in 2021, which has led to the complete burial of urban facilities, buildings and roads.

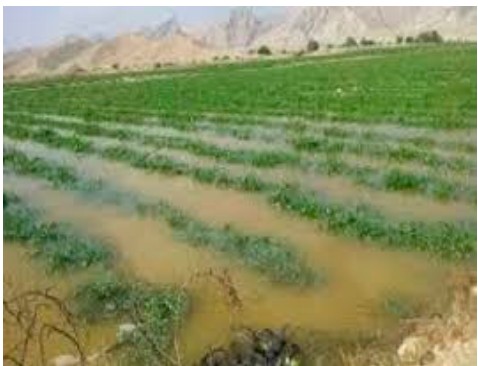
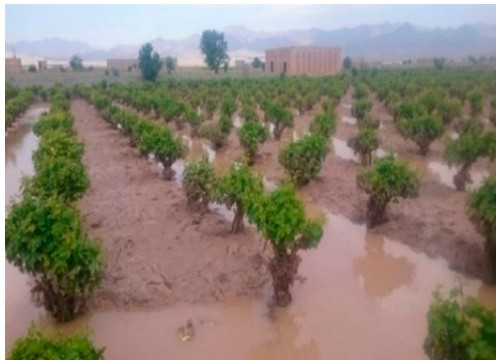

**Figure 3.** The water retention in the southern parts of the studied region with agricultural use due to 3-day rainfall in rainy seasons in 2021.

These aforementioned conditions show that the daily rains have been able to destroy most of the facilities and agricultural lands and gardens. In this research, the characteristics of the studied area were identified. The correct view of the potential of underground and surface water in the area was obtained. Finally, by implementing optimal distances between underground drains, the challenge created in this area can be solved.

Throughout history, drainage systems have generally been used to remove excess water from the ground surface. However, during the last two decades, a more comprehensive approach to drainage plans has emerged and states that drainage is one of the most important parameters for the development and management of water resources [15]. Drainage is considered an important parameter in the efficiency of agricultural lands in dry areas, but it is often neglected as an important source of water supply. Although the design and management of irrigation systems have improved greatly in the last few decades, the

design method and criteria for subsurface drainage management have not changed in the last 50 years. Therefore, all aspects of agricultural drainage should be seriously reviewed and re-evaluated to meet the emerging needs of today's world, which is continuously dealing with water scarcity problems [16].

In 2020, Moshayedi et al. [17] used the drain spacing, the steady Hooghoudt model, and the unsteady Glover–Dumm, modified Glover and van Schilfgaarde models in rainy periods of 2, 3, 4, and 5 days, and the distance between computational drains at two depths to install drains (180 cm and 200 cm) was compared with the results of the subsurface drainage project in the lands of the Senejan plain. To compare the results, the mean absolute error (MAE) and standard error σ were used. The results showed that MAE and σ values in the 3-, 4-, and 5-day rainfall in the Glover-Dumm model are 9.75 and 0.196% for the depth of 180 cm and 7.944, and 0.15% for the depth of 200 cm, respectively. The above coefficients show 8.098 and 0.162% for the depth of 180 cm and 6.6% and 6.685 for the depth of 200 cm in 2-day precipitation, respectively, in the van Schilfgarde model. Additionally, drainable pore space μ is considered one of the main criteria in the design of underground drainage systems in unsteady flow conditions. The μ parameter is generally a time-consuming and difficult process to determine. This parameter is determined by the soil moisture conditions in one situation and time. When the depth of groundwater is relatively low, the value of this parameter varies greatly in time and place. For this reason, when designing underground drainage systems, special attention should be given to them.

Darzi Naftchali et al. [18] stated in the year 2014 that in order to calculate the distance between the drains, the time parameter has an effect on the equations for determining the distances between the drains. They added that in time-independent conditions, Hogwood, Kirkham, Dagan, Ernst, and Ernst Hogood equations can be used, and in time-dependent conditions, Glover-Damm, Bower, Van Schilfgaarde, and Dezio equations can be used. Mardokhi et al. [19] considered the determination of the coefficient of hydraulic head as one of the most important factors in determining the distances between drains. By drilling 30 research wells in Khuzestan region, they stated that the reverse well method with a correlation coefficient of 0.51 is the optimal method in determining the hydraulic conductivity coefficient.

Karimi et al. [20] stated that the design and operation of a subsurface drainage system largely depended on the saturation discharge coefficient. He used hole tests, inverted drill holes, and piezometers for evaluation. These methods are based on the fact that the water flow is determined in a certain volume of soil, its boundary conditions are determined, and the amount of hydraulic conductivity of soil saturation is evaluated using the relationships obtained from Darcy's law. Dehghani et al. [21] determined the hydraulic conductivity of the soil to design a subsurface drainage network on 200 square meters of land in the Weiss area of Ahvaz by drilling 30 holes and using a golf penetration meter as a modern method and hole pumping method. He expressed the best relationship between hydraulic conductivity obtained from both methods, an exponential relationship with a correlation coefficient of 0.28. In 2013, Ali Pour et al. [22] showed that the inverse auger hole method presented the hydraulic conductivity beyond the Guelph permeameter method by drilling 20 holes at an area of 2000 ha of Hoveyzeh town and using the inverse auger hole method and Guelph permeameter, and also stated that the best relationship between the hydraulic conductivity obtained from both methods is a linear relationship with a correlation coefficient of 0.75. Rahimijomnani et al. [23] evaluated the installation of underground drains at a depth of 2 m from the ground in northern Ahvaz, reduced the depth of the drains by 1.2 m, and re-evaluated them. The results showed that the underground discharge distance, due to changes in hydraulic conductivity measured at a depth of 2 m, was 30% less than previous studies measured at a depth of 3 m. Rafiee Niya et al. [24–34] evaluated the primary parameters influencing the design of subsurface drains in his experimental farm in Mino Island. He stated that the physical and chemical properties of the soil change from one point to another and will affect the distance between the drains. Some other researchers have worked in this area including [25–32].

The results of previous research indicate that the implementation of drainage projects and the proper conduction and transfer of surface and subsurface waters, as well as the identification of effective parameters in the process of drainage projects, are very important. In general, the formation of any type of uncontrolled runoff and its flow at the surface of facilities can cause great damage and threaten the health of any project. Previous studies have paid less attention to comparing measured numerical values with computational numerical values obtained from numerical models in unsteady conditions. Additionally, in fewer studies, the rock bed has been dominant in the land and calculations, and the existence of this rock bed has had a direct impact on the depth of drilling and execution speed. According to the previous findings, the hydraulic conductivity of the soil can be evaluated using the inverse auger hole, which can provide good numerical results. Previous results showed that in drainage calculations and the determination of optimal values, the soil type, impermeable layer depth, and meteorological and climatic parameters are effective.

## 3. Materials and Methods

The studied region is located in the center of Iran in Markazi Province, with an area of over 70 hectares at the westernmost point of the city of Arak consisting of the mountain in the northern part and the plain in the southern part. The altitude of this area is 1780 m (above sea level), and based on the meteorological data for the years 1967 to 2021 which was prepared by the Meteorology Department of the study area located in the synoptic station, the average annual rainfall of the study area is equal to 319.57 mm, and the average monthly temperature of the study area is reported as 13.88 degrees Celsius. Additionally, the climate of the region is cold and dry based on meteorological findings, so from the beginning of June to the middle of November is defined as the dry period, and from the middle of late November to the beginning of May as the wet period which can be seen in the Ombrothermic diagram shown in Figure 4. Among the most important parameters that influence the numerical calculation of the drain spacing is the identification of the saturated hydraulic conductivity of the soil. For this purpose, computational or laboratory numerical models can be used. Some of the computational models are presented in Table 1. In these relations, the drain spacing is presented with the symbol (L) in meters, the standard depth of the depth of drain placement is expressed with the symbol ($h_0$) in meters, and hydraulic conductivity is presented with the symbol (K).

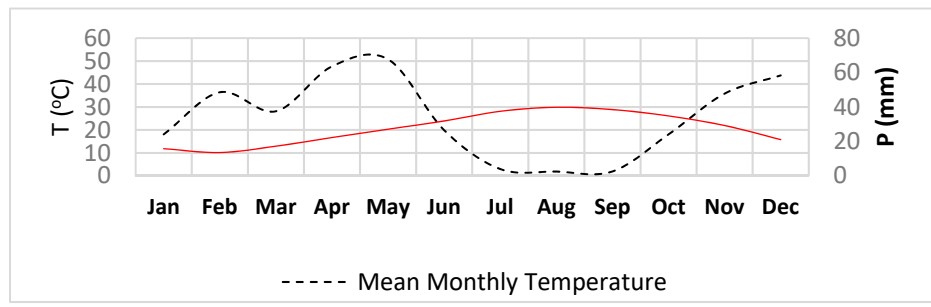

**Figure 4.** Ombrothermic diagram based on the meteorological data (1996–2016) at the synoptic station of Arak.

In this study, the researcher used the inverse auger hole method by conducting field research and identifying flood susceptibility points in the region, which have been identified during the last ten years and have caused the most damage due to 1- to 5-day rainfall. According to the findings of previous researchers, the inverse auger hole method is one of the most widely used and simplest methods for determining the hydraulic conductivity of saturated soil. For this purpose, the district was divided into 44 research blocks, and 22 boreholes were drilled in these blocks. The dimensions of each borehole were considered to be 80 cm in depth and 10 cm in diameter, and all boreholes were fed manually with

water. The results of water level fall due to leakage from the hole wall at alternating times of 2 min were recorded, and the value of hydraulic conductivity ($k$) according to Table 2 was obtained by drawing a semi logarithmic graph for each borehole. To determine the parameter k, the values of the parameters $Q_1$ and $Q_2$, the depth of water ($h$), and the radius of the hole ($r$) were defined according to Equation (1) [33].

$$Q_2 = -\pi r^2 - \frac{dh}{dt} \tag{1}$$

$$Q_1 = K\left(2\pi rh + \pi r^2\right) = 2\pi kr\left(h + \frac{r}{2}\right)$$

$$Q_1 = Q_2$$

$$Q = \frac{2K}{r}dt = -\frac{dh}{h + r/2}$$

$$K = 1.15\, r\, \frac{\left[\log\left(h_0 + \frac{r}{2}\right) - \log\left(h_t + \frac{r}{2}\right)\right]}{t} = 1.15\, r\, \frac{\log\left[h(t_1) + \frac{r}{2}\right] - \log\left[h(t_n) + \frac{r}{2}\right]}{t_n - t_1} = 1.15\, r\, \tan\alpha$$

**Table 1.** Practical relationships for calculating hydraulic conductivity ($k$) in unsteady conditions and inverse auger hole laboratory model [33].

| Practical Relation | Numerical Study Model |
|---|---|
| Dumm Glover | $K = \frac{\frac{L^2 q_t}{h_t}}{2\pi D}$ |
| Hemmad | $K = \frac{\left(\frac{q_t}{h_t}\right) L \ln\left(\frac{L^2}{2\pi^2 \mathrm{rd}}\right)}{2\pi}$ |
| Van Schilfgaarde | $K = \frac{2L^2\left(\frac{q_t}{h_t}\right)}{9(2d_e + h_0)}$ |
| Bouwer & Van Schilfgarrde | $K = \frac{L^2\left(\frac{q_t}{h_t}\right)}{4(2d_e + h_0)}$ |
| Auger hole method | $K = 1.15\, r \tan\alpha$ |

**Table 2.** Numerical values of soil hydraulic conductivity ($k$) in 22 agrology boreholes ($d$) using the inverse auger hole laboratory model.

| $d$ | $K$ | $d$ | $K$ |
|---|---|---|---|
| 1 | 0.795 | 12 | 0.596 |
| 2 | 0.596 | 13 | 0.795 |
| 3 | 0.596 | 14 | 0.795 |
| 4 | 0.596 | 15 | 0.596 |
| 5 | 0.795 | 16 | 0.795 |
| 6 | 0.596 | 17 | 0.795 |
| 7 | 0.596 | 18 | 0.795 |
| 8 | 0.795 | 19 | 0.795 |
| 9 | 0.596 | 20 | 0.795 |
| 10 | 0.596 | 21 | 0.795 |
| 11 | 0.596 | 22 | 0.795 |

To identify the type of soil in each research block of the region, the soil collected from all research boreholes was transferred to the laboratory for soil grain size testing, and after determining the coefficients of soil softness and drawing the soil grain size diagram, the soil type in each borehole was evaluated according to Table 3. The effective parameters in this table include the classified blocks of land ($b$), the area of each block in square meters ($s$), the number of study boreholes ($d$), and the type of soil in each borehole ($t$).

**Table 3.** Numerical results obtained for the area of each research block (*b*) in terms of square meters (*s*), number of study boreholes (*d*), and soil type in each research borehole (*t*).

| *t* | *d* | *s* | *b* | *t* | *d* | *s* | *b* |
|---|---|---|---|---|---|---|---|
| GC | 34–50 | 6051.11 | 23 | GW | 15 | 11,614.68 | 1 |
| GC | 35 | 25,992.26 | 24 | SC | 16 | 8924.52 | 2 |
| GM-GC-GC | 10–11–12 | 27,365.27 | 25 | SC | 22 | 8122.71 | 3 |
| GC | 7 | 12,157.96 | 26 | SC | 21 | 6494.13 | 4 |
| GC | 41 | 11,951.67 | 27 | SC | 18 | 2893.8 | 5 |
| GC | 42 | 9836.43 | 28 | SC | 19 | 4271.94 | 6 |
| GC-GM | 2–9 | 16,771.18 | 29 | SC | 20 | 13,104.62 | 7 |
| GM | 6 | 11,091.26 | 30 | GC | 21 | 12,575.75 | 8 |
| GM | 4 | 17,787.80 | 31 | GC | 22 | 11,635.48 | 9 |
| GM | 3 | 23,200.77 | 32 | GC | 23 | 3374.63 | 10 |
| CL | 40 | 8913.84 | 33 | GC | 28 | 4565.15 | 11 |
| CL | 8 | 10,353.99 | 34 | GC | 29 | 2498.23 | 12 |
| CL | 1 | 12,823.66 | 35 | GC | 24 | 1276.9 | 13 |
| SC | 5 | 5859.6 | 36 | GC | 30 | 8547.28 | 14 |
| SC | 38–49 | 27,184.81 | 37 | SC | 31 | 8183.47 | 15 |
| SC | 37–48 | 54,228.83 | 38 | SC | 25 | 10,561.25 | 16 |
| GM | 39 | 15,777.17 | 39 | SC | 32 | 5369.74 | 17 |
| GM | 44 | 20,923.78 | 40 | SC | 26 | 11,672.63 | 18 |
| GM | 36 | 5441.3 | 41 | SC | 27 | 5754.96 | 19 |
| GM | 46–47 | 79,925.02 | 42 | SM | 14 | 10,390.87 | 20 |
| GM | 43 | 16,866.26 | 43 | SM | 33 | 9944.25 | 21 |
| GM | 45 | 14,284.55 | 44 | SM | 17 | 10,492.21 | 22 |

The initial parameters for determining the drain spacing were calculated according to the previously presented tables. Time-dependent or so-called nonsteady conditions have been used to determine the drain spacing. Drain spacing has been considered 50 m in the area according to conducted studies, and daily rainfall over the past 10 years and the formation of runoff from rainfall in the northern region and its conduction to the southern part with agricultural lands have shown that the performance of the implemented drainage project is undesirable and it cannot properly conduct and transfer excess water during seasonal rainfall. Computational studies for determining the drain spacing were examined at variable depths of 110 cm to a maximum depth of 180 cm using 5 numerical models, including Dumm, Glover, Bouwer, Hemmad, and Bouwer & Van Schilfgaarde, according to Equations (2)–(6) and the results of Table 4 was investigated in nonsteady conditions. Parameters (*m*) and ($m_0$) indicate the placement of drains in the primary and secondary depths in meters. These two different depths, due to the presence of a rock bed, are intended to reduce the volume of excavation and reduce drilling costs. *k* indicates the hydraulic conductivity of saturated soil. Additionally, in all numerical models, due to the sensitivity of the subject and the high potential of the region in the formation of seasonal runoff, the duration of rainfall is considered to be in the range of 1 to 5 days.

$$\text{Dumm} = L^2 = \frac{\pi^2 K D_t}{f \ln \frac{4m_0}{\pi m}} \tag{2}$$

$$\text{Glover} = L = \left| \frac{\frac{KDT}{P}}{\ln \frac{4z_0}{\pi z_T}} \right|^{\frac{1}{2}} \tag{3}$$

$$\text{Hemmad} = L = \frac{2\pi K_t}{f \ln \left( \frac{m_0}{m} \right) \ln \left( \frac{L^2}{2\pi^2 rd} \right)} \tag{4}$$

$$\text{Bouwer} = L^2 = \frac{9 \, k t_d}{f \ln \frac{m_0(m+2\,d)}{m(m_0+2\,d)}} \tag{5}$$

$$\text{Bouwer \& Schilfgarrde} = L^2 = \frac{8\,kt_d}{Cf\,\ln\frac{m_0(m+2\,d)}{m(m_0+2\,d)}} \tag{6}$$

**Table 4.** Distances between computational drains based on numerical elationships in time-dependent conditions at a depth of 110 cm; 18 cm above the ground and under a rainfall of 1–5 days.

| Depth of Drainage $h$ (cm) | 1-Day Rainfall | 2-Day Rainfall | 3-Day Rainfall | 4-Day Rainfall | 5-Day Rainfall |
|---|---|---|---|---|---|
| Drain spacing in numerical model Dumm (m) | | | | | |
| $h = 110$ cm | 12.02 | 12.53 | 14.01 | 17.01 | 36.67 |
| $h = 120$ cm | 13.76 | 14.47 | 16.51 | 20.41 | 26.6 |
| $h = 130$ cm | 15.06 | 15.91 | 18.34 | 22.68 | 28.81 |
| $h = 140$ cm | 16.13 | 17.09 | 19.79 | 24.37 | 30.22 |
| $h = 150$ cm | 17.05 | 18.11 | 21 | 25.68 | 31.2 |
| $h = 160$ cm | 17.87 | 19 | 22.03 | 26.73 | 31.92 |
| $h = 170$ cm | 18.61 | 19.8 | 22.93 | 27.6 | 32.48 |
| $h = 180$ cm | 19.28 | 20.51 | 23.72 | 28.33 | 32.92 |
| Drain spacing in numerical model Glover (m) | | | | | |
| $h = 110$ cm | 9.59 | 14.28 | 19.89 | 28.19 | 68.73 |
| $h = 120$ cm | 10.77 | 16.17 | 23 | 33.2 | 67.4 |
| $h = 130$ cm | 10.77 | 16.17 | 23 | 33.2 | 67.4 |
| $h = 140$ cm | 11.54 | 17.43 | 25.02 | 36.17 | 66.03 |
| $h = 150$ cm | 12.51 | 18.99 | 27.43 | 38.04 | 64.63 |
| $h = 160$ cm | 12.8 | 19.45 | 27.43 | 39.19 | 61.7 |
| $h = 170$ cm | 13 | 19.76 | 28.52 | 40.11 | 60.16 |
| $h = 180$ cm | 13.11 | 19.93 | 28.72 | 40.08 | 58.56 |
| Drain spacing in numerical model Hemmad (m) | | | | | |
| $h = 110$ cm | 3.52 | 4.83 | 5.79 | 7.1 | 11.23 |
| $h = 120$ cm | 4.28 | 5.82 | 7.72 | 9.88 | 11.92 |
| $h = 130$ cm | 4.75 | 6.41 | 8.35 | 10.4 | 12.19 |
| $h = 140$ cm | 5.08 | 6.82 | 8.75 | 10.69 | 12.34 |
| $h = 150$ cm | 5.34 | 7.12 | 9.03 | 10.89 | 12.43 |
| $h = 160$ cm | 5.53 | 7.36 | 9.25 | 11.04 | 12.49 |
| $h = 170$ cm | 5.7 | 7.54 | 9.42 | 11.14 | 12.53 |
| $h = 180$ cm | 5.83 | 7.7 | 9.55 | 11.22 | 12.57 |
| Drain spacing in numerical model Bouwer (m) | | | | | |
| $h = 110$ cm | 5.64 | 8.42 | 11.87 | 17.46 | 29.2 |
| $h = 120$ cm | 6.62 | 10 | 14.55 | 26.33 | 47.46 |
| $h = 130$ cm | 7.4 | 11.27 | 16.74 | 26.52 | 47.85 |
| $h = 140$ cm | 8.09 | 12.4 | 18.68 | 30.1 | 55.06 |
| $h = 150$ cm | 8.72 | 13.43 | 20.45 | 33.36 | 61.59 |
| $h = 160$ cm | 9.31 | 14.39 | 22.11 | 36.39 | 67.62 |
| $h = 170$ cm | 9.87 | 15.31 | 23.68 | 39.25 | 73.28 |
| $h = 180$ cm | 10.41 | 16.18 | 25.18 | 41.96 | 78.64 |
| Drain spacing in numerical model Bouwer & Van Schilfgarrde (m) | | | | | |
| $h = 110$ cm | 6.04 | 9.01 | 12.68 | 18.61 | 31.08 |
| $h = 120$ cm | 7.12 | 10.74 | 15.6 | 24.01 | 42.23 |
| $h = 130$ cm | 7.99 | 12.16 | 18 | 28.45 | 51.26 |
| $h = 140$ cm | 8.77 | 13.41 | 20.14 | 32.38 | 59.17 |
| $h = 150$ cm | 9.48 | 14.57 | 22.12 | 36 | 66.37 |
| $h = 160$ cm | 10.14 | 15.65 | 23.98 | 39.37 | 73.07 |
| $h = 170$ cm | 10.78 | 16.69 | 25.74 | 42.57 | 79.4 |
| $h = 180$ cm | 11.4 | 17.69 | 27.44 | 45.64 | 85.44 |

To better compare the results of the numerical drainage models used in this study, the mean absolute error (MAE) according to Equation (7), standard error σ according to Equation (8), and reaction modulus α of Equation (9) were used. In the σ and MAE relationships, parameter β1 is equal to the computational drain distance in the region in m, β2 is equal to the distance of the measured drains in m, and n is the number of drilled boreholes. The numerical results of parameters σ and MAE are presented in Table 5.

**Table 5.** Mean numerical percentage (σ) and (MAE) under the numerical model of unsteady conditions and daily rainfall in the depth of drain (h) of 110 cm; 180 cm.

| Day Rainfall | h = 110 cm | | h = 120 cm | | h = 130 cm | | h = 140 cm | | h = 150 cm | | h = 160 cm | | h = 170 cm | | h = 180 cm | |
|---|---|---|---|---|---|---|---|---|---|---|---|---|---|---|---|---|
| | σ | MAE | σ | MAE | σ | MAE | σ | MAE | σ | MAE | σ | MAE | σ | MAE | σ | MAE |
| Numerical values of σ and MAE in the Dumm numerical model | | | | | | | | | | | | | | | | |
| 1-day Rainfall | 0.76 | 38.03 | 0.73 | 36.36 | 0.7 | 34.99 | 0.68 | 33.92 | 0.66 | 32.99 | 0.64 | 32.17 | 0.63 | 31.43 | 0.62 | 30.76 |
| 2-day Rainfall | 0.75 | 37.57 | 0.71 | 35.71 | 0.68 | 34.18 | 0.66 | 33 | 0.64 | 31.98 | 0.62 | 31.09 | 0.61 | 30.3 | 0.59 | 29.58 |
| 3-day Rainfall | 0.72 | 36.12 | 0.67 | 33.74 | 0.64 | 31.8 | 0.61 | 30.35 | 0.58 | 29.14 | 0.56 | 28.11 | 0.54 | 27.21 | 0.53 | 26.42 |
| 4-day Rainfall | 0.66 | 33.18 | 0.6 | 29.99 | 0.55 | 27.5 | 0.52 | 25.82 | 0.49 | 24.51 | 0.47 | 23.45 | 0.45 | 22.59 | 0.44 | 21.86 |
| 5-day Rainfall | 0.27 | 27.19 | 0.47 | 24.1 | 0.43 | 21.42 | 0.4 | 20.01 | 0.38 | 19.03 | 0.37 | 18.3 | 0.36 | 17.75 | 0.35 | 17.3 |
| Numerical values of σ and MAE in the Glover numerical model | | | | | | | | | | | | | | | | |
| 1-day Rainfall | 0.81 | 40.45 | 0.79 | 39.38 | 0.79 | 39.28 | 0.77 | 38.5 | 0.75 | 37.54 | 0.75 | 37.24 | 0.74 | 37.05 | 0.74 | 36.94 |
| 2-day Rainfall | 0.72 | 35.81 | 0.68 | 34.17 | 0.68 | 33.92 | 0.65 | 32.66 | 0.62 | 31.1 | 0.61 | 30.64 | 0.61 | 30.33 | 0.6 | 30.16 |
| 3-day Rainfall | 0.61 | 30.24 | 0.54 | 27.77 | 0.54 | 27.14 | 0.5 | 25.11 | 0.45 | 22.7 | 0.45 | 22.7 | 0.43 | 21.62 | 0.43 | 21.42 |
| 4-day Rainfall | 0.44 | 21.99 | 0.34 | 19.01 | 0.34 | 16.99 | 0.28 | 14.16 | 0.24 | 13.4 | 0.22 | 13.27 | 0.2 | 13.93 | 0.2 | 14.45 |
| 5-day Rainfall | −0.37 | 19.62 | −0.34 | 24.41 | −0.34 | 19.04 | −0.32 | 18.49 | −0.29 | 17.95 | −0.23 | 17.41 | −0.20 | 17.56 | −0.17 | 17.86 |
| Numerical values of σ and MAE in the Hemmad numerical model | | | | | | | | | | | | | | | | |
| 1-day Rainfall | 0.93 | 46.52 | 0.92 | 45.77 | 0.91 | 45.29 | 0.9 | 44.96 | 0.89 | 44.71 | 0.89 | 44.51 | 0.89 | 44.35 | 0.88 | 44.21 |
| 2-day Rainfall | 0.91 | 45.26 | 0.89 | 44.28 | 0.87 | 43.68 | 0.87 | 43.27 | 0.86 | 42.97 | 0.86 | 42.74 | 0.85 | 42.55 | 0.85 | 42.39 |
| 3-day Rainfall | 0.89 | 44.34 | 0.85 | 42.43 | 0.84 | 41.79 | 0.83 | 41.39 | 0.82 | 41.1 | 0.82 | 40.89 | 0.81 | 40.72 | 0.81 | 40.59 |
| 4-day Rainfall | 0.86 | 43.08 | 0.81 | 40.32 | 0.8 | 39.79 | 0.79 | 39.49 | 0.79 | 39.29 | 0.78 | 39.15 | 0.78 | 39.04 | 0.78 | 38.96 |
| 5-day Rainfall | 0.78 | 39 | 0.77 | 38.34 | 0.76 | 38.04 | 0.76 | 37.89 | 0.76 | 37.8 | 0.76 | 37.74 | 0.75 | 37.69 | 0.75 | 37.66 |
| Numerical values of σ and MAE in the Bouwer numerical model | | | | | | | | | | | | | | | | |
| 1-day Rainfall | 0.89 | 44.41 | 0.87 | 43.43 | 0.85 | 42.64 | 0.84 | 41.95 | 0.83 | 41.32 | 0.82 | 40.73 | 0.8 | 40.17 | 0.79 | 39.64 |
| 2-day Rainfall | 0.83 | 41.67 | 0.8 | 40.09 | 0.78 | 38.82 | 0.75 | 37.69 | 0.73 | 36.66 | 0.71 | 35.7 | 0.7 | 34.78 | 0.68 | 33.91 |
| 3-day Rainfall | 0.83 | 38.26 | 0.71 | 35.59 | 0.67 | 33.39 | 0.63 | 31.46 | 0.59 | 29.68 | 0.56 | 28.03 | 0.53 | 26.46 | 0.5 | 24.96 |
| 4-day Rainfall | 0.65 | 32.72 | 0.48 | 23.88 | 0.47 | 23.66 | 0.4 | 20.08 | 0.34 | 16.82 | 0.28 | 13.79 | 0.22 | 10.94 | 0.16 | 8.22 |
| 5-day Rainfall | 0.42 | 21.03 | 0.06 | 3.52 | 0.05 | 2.38 | −0.10 | 4.84 | −0.23 | 11.36 | −0.35 | 17.39 | −0.46 | 23.05 | −0.57 | 28.42 |
| Numerical values of σ and MAE in the Bouwer & Van Schilfgarrde numerical model | | | | | | | | | | | | | | | | |
| 1-day Rainfall | 0.88 | 44 | 0.86 | 42.92 | 0.84 | 42.05 | 0.83 | 41.28 | 0.81 | 40.57 | 0.8 | 39.9 | 0.79 | 39.26 | 0.77 | 36.65 |
| 2-day Rainfall | 0.82 | 41.08 | 0.79 | 39.35 | 0.76 | 37.93 | 0.73 | 36.68 | 0.71 | 35.52 | 0.69 | 34.44 | 0.67 | 33.4 | 0.65 | 32.4 |
| 3-day Rainfall | 0.75 | 37.46 | 0.69 | 34.54 | 0.64 | 32.14 | 0.6 | 29.99 | 0.56 | 28.02 | 0.52 | 26.16 | 0.49 | 24.39 | 0.45 | 22.7 |
| 4-day Rainfall | 0.63 | 31.57 | 0.52 | 26.19 | 0.44 | 21.73 | 0.36 | 17.8 | 0.28 | 14.19 | 0.22 | 10.81 | 0.15 | 7.61 | 0.09 | 4.54 |
| 5-day Rainfall | 0.38 | 19.15 | 0.16 | 8.13 | −0.02 | 1.78 | −0.18 | 8.95 | −0.32 | 16.15 | −0.46 | 22.85 | −0.58 | 29.18 | −0.70 | 35.22 |

Numerical RMSE values for each of the numerical models are also presented in Table 6. Equation (9) was used to estimate the reaction modulus α, which indicates the behavior or changes in the drainage intensity, and the results are presented in Table 7. The value of this parameter can be variable for different lands according to the change in KD-L-μ coefficients, and the results are presented in Table 8.

$$MAE_{all} = \frac{1}{n}\sum |(\beta_1) - (\beta_2)| \tag{7}$$

$$\sigma = \frac{(\beta_1) - (\beta_2)}{(\beta_1)} \times 100\% \tag{8}$$

$$\alpha = \frac{\ln h_{t-1} - \ln h_t}{\Delta t} = 2.30\,\frac{\log h_{t-1} - \log h_t}{\Delta t} = \frac{10\,KD}{\mu\,L^2} \tag{9}$$

**Table 6.** Numerical values of (RMSE) under daily rainfall for 1 to 5 days in the depth of drainage at the rate of 110 cm; 180 cm above the ground.

| Numerical Model | $h$ = 110 cm | $h$ = 120 cm | $h$ = 130 cm | $h$ = 140 cm | $h$ = 150 cm | $h$ = 160 cm | $h$ = 170 cm | $h$ = 180 cm |
|---|---|---|---|---|---|---|---|---|
| | | | | 1-Day rainfall | | | | |
| Hemmad | 46.524 | 45.765 | 45.297 | 44.966 | 44.715 | 44.517 | 44.355 | 44.219 |
| Bouwer & Van | 44.004 | 42.924 | 42.052 | 41.28 | 40.569 | 39.9 | 39.263 | 38.65 |
| Bouwer | 44.411 | 43.43 | 42.645 | 41.955 | 41.324 | 40.735 | 40.175 | 39.64 |
| Dumm | 38.078 | 36.355 | 35.074 | 34.018 | 33.109 | 32.305 | 31.583 | 29.93 |
| Glover | 40.524 | 39.376 | 39.376 | 38.625 | 37.711 | 37.447 | 37.281 | 37.202 |
| | | | | 2-Day rainfall | | | | |
| Hemmad | 45.263 | 44.277 | 43.688 | 43.281 | 42.979 | 42.744 | 42.555 | 42.399 |
| Bouwer & Van | 41.08 | 39.347 | 37.936 | 36.683 | 35.528 | 34.441 | 33.404 | 32.407 |
| Bouwer | 41.674 | 40.093 | 38.817 | 37.693 | 36.663 | 35.7 | 34.787 | 33.913 |
| Dumm | 37.623 | 35.707 | 34.285 | 33.121 | 32.125 | 31.253 | 30.476 | 29.17 |
| Glover | 35.985 | 34.17 | 34.17 | 32.991 | 31.588 | 31.202 | 30.976 | 30.89 |
| | | | | 3-Day rainfall | | | | |
| Hemmad | 44.347 | 42.428 | 41.803 | 41.399 | 41.114 | 40.901 | 40.736 | 40.604 |
| Bouwer & Van | 37.462 | 34.543 | 32.141 | 30.001 | 28.029 | 26.175 | 24.411 | 22.71 |
| Bouwer | 38.266 | 35.585 | 33.396 | 31.462 | 29.691 | 28.037 | 26.471 | 24.977 |
| Dumm | 36.202 | 33.741 | 31.956 | 30.538 | 29.362 | 28.363 | 27.497 | 26.739 |
| Glover | 30.641 | 27.772 | 27.772 | 25.984 | 24.059 | 24.059 | 23.45 | 23.487 |
| | | | | 4-Day rainfall | | | | |
| Hemmad | 43.089 | 40.315 | 39.806 | 39.5 | 39.309 | 39.169 | 39.064 | 38.983 |
| Bouwer & Van | 31.578 | 26.187 | 21.754 | 17.83 | 14.24 | 10.893 | 7.748 | 4.81 |
| Bouwer | 32.723 | 23.875 | 23.676 | 20.105 | 16.858 | 13.84 | 11.018 | 8.345 |
| Dumm | 33.311 | 29.994 | 27.789 | 26.172 | 24.923 | 23.925 | 23.107 | 22.424 |
| Glover | 23.072 | 19.008 | 19.008 | 17.021 | 16.108 | 15.82 | 16.277 | 16.827 |
| | | | | 5-Day rainfall | | | | |
| Hemmad | 39.021 | 38.335 | 38.065 | 37.919 | 37.829 | 37.767 | 37.722 | 37.688 |
| Bouwer & Van | 19.181 | 8.126 | 2.02 | 9.167 | 16.302 | 22.982 | 29.304 | 35.342 |
| Bouwer | 21.057 | 3.516 | 2.887 | 5.189 | 11.552 | 17.541 | 23.183 | 28.543 |
| Dumm | 15.081 | 24.098 | 22.031 | 20.728 | 19.831 | 19.174 | 18.673 | 18.278 |
| Glover | 25.169 | 24.414 | 24.414 | 23.706 | 23.057 | 21.993 | 21.614 | 21.368 |

**Table 7.** Numerical values of reaction modulus ($\alpha$) in numerical models of unsteady conditions and depth install drain $h$ under 1- to 5-day rainfall.

| Depth of Drainage $h$ (cm) | Numerical Model Hemmad | Numerical Model Bouwer & Van | Numerical Model Bouwer | Numerical Model Dumm | Numerical Model Glover |
|---|---|---|---|---|---|
| | | | 1-Day rainfall | | |
| $h$ = 110 cm | 26.676 | 9.355 | 10.761 | 2.263 | 3.725 |
| $h$ = 120 cm | 17.964 | 6.731 | 7.802 | 1.725 | 2.977 |
| $h$ = 130 cm | 14.566 | 5.344 | 6.235 | 1.439 | 2.977 |
| $h$ = 140 cm | 12.712 | 4.447 | 5.22 | 1.254 | 2.613 |
| $h$ = 150 cm | 11.531 | 3.808 | 4.495 | 1.121 | 2.28 |
| $h$ = 160 cm | 10.71 | 3.325 | 3.946 | 1.021 | 2.215 |
| $h$ = 170 cm | 10.103 | 2.946 | 3.514 | 0.941 | 2.194 |
| $h$ = 180 cm | 9.636 | 2.64 | 3.165 | 0.877 | 2.216 |
| | | | 2-Day rainfall | | |
| $h$ = 110 cm | 28.709 | 8.449 | 9.691 | 4.2 | 3.372 |
| $h$ = 120 cm | 19.659 | 5.933 | 6.856 | 3.143 | 3.143 |
| $h$ = 130 cm | 16.158 | 4.635 | 5.389 | 2.596 | 2.646 |
| $h$ = 140 cm | 14.258 | 3.809 | 4.456 | 2.246 | 2.298 |
| $h$ = 150 cm | 13.055 | 3.23 | 3.8 | 2 | 1.984 |
| $h$ = 160 cm | 12.222 | 2.798 | 3.309 | 1.816 | 1.922 |
| $h$ = 170 cm | 11.609 | 2.463 | 2.928 | 1.673 | 1.902 |
| $h$ = 180 cm | 11.139 | 2.195 | 2.622 | 1.557 | 1.92 |

**Table 7.** *Cont.*

| Depth of Drainage *h* (cm) | Numerical Model Hemmad | Numerical Model Bouwer & Van | Numerical Model Bouwer | Numerical Model Dumm | Numerical Model Glover |
|---|---|---|---|---|---|
| 3-Day rainfall | | | | | |
| *h* = 110 cm | 23.378 | 6.404 | 7.31 | 5.067 | 2.609 |
| *h* = 120 cm | 16.836 | 4.221 | 4.85 | 3.638 | 1.964 |
| *h* = 130 cm | 14.362 | 3.166 | 3.661 | 2.946 | 1.964 |
| *h* = 140 cm | 13.042 | 2.529 | 2.94 | 2.526 | 1.673 |
| *h* = 150 cm | 12.217 | 2.098 | 2.453 | 2.242 | 1.426 |
| *h* = 160 cm | 11.652 | 1.786 | 2.1 | 2.036 | 1.426 |
| *h* = 170 cm | 11.239 | 1.55 | 1.832 | 1.879 | 1.37 |
| *h* = 180 cm | 10.925 | 1.366 | 1.622 | 1.755 | 1.388 |
| 4-Day rainfall | | | | | |
| *h* = 110 cm | 17.389 | 3.95 | 4.49 | 4.603 | 1.731 |
| *h* = 120 cm | 17.389 | 3.95 | 4.49 | 4.603 | 1.731 |
| *h* = 130 cm | 13.709 | 2.367 | 1.973 | 3.186 | 1.256 |
| *h* = 140 cm | 12.37 | 1.684 | 1.938 | 2.573 | 1.256 |
| *h* = 150 cm | 11.245 | 1.052 | 1.225 | 2.005 | 0.975 |
| *h* = 160 cm | 10.954 | 0.88 | 1.03 | 1.849 | 0.931 |
| *h* = 170 cm | 10.745 | 0.753 | 0.886 | 1.734 | 0.923 |
| *h* = 180 cm | 10.586 | 0.656 | 0.776 | 1.645 | 0.914 |
| 5-Day rainfall | | | | | |
| *h* = 110 cm | 13.325 | 1.761 | 1.997 | 1.228 | 0.362 |
| *h* = 120 cm | 11.8 | 0.951 | 0.757 | 2.343 | 0.379 |
| *h* = 130 cm | 11.27 | 0.65 | 0.74 | 1.995 | 0.379 |
| *h* = 140 cm | 11 | 0.485 | 0.559 | 1.812 | 0.399 |
| *h* = 150 cm | 10.836 | 0.385 | 0.447 | 1.699 | 0.421 |
| *h* = 160 cm | 10.727 | 0.318 | 0.371 | 1.622 | 0.477 |
| *h* = 170 cm | 10.648 | 0.27 | 0.317 | 1.567 | 0.512 |
| *h* = 180 cm | 10.589 | 0.233 | 0.275 | 1.524 | 0.555 |

**Table 8.** Fluctuation range of parameter $\alpha$ in terms of numerical variations KD-L-μ.

| Range ($\alpha$) | μ | L | KD |
|---|---|---|---|
| $0.3 < \alpha < 0.2$ | high | high | low |
| $2 < \alpha < 5$ | low | low | high |

Numerical results obtained from RMSE values according to Table 6 show that the minimum values in each numerical model and per day of continuous rainfall belong to Dumm at a depth of 180 cm from the surface and under 1-day rainfall, Dumm at a depth of 180 cm from the surface and under 2-day rainfall, Bouwer & Van at a depth of 180 cm from the surface and under 3-day rainfall, Dumm at a depth of 180 cm from the surface and under 4-day rainfall, and Dumm at a depth of 130 cm from the surface and under 5-day rainfall.

## 4. Conclusions

In this research, according to the diversity of underground layers at different depths, the distance of subsurface drains was calculated using the numerical equations of Bouwer & Van Schilfgarrd, Dumm, Glover, Hemmad, and Bouwer. The range of the distance between the drains was 35.64 m to 85.4 m. In order to choose the best computational model, the calculated values were compared with the measured values in the area. Additionally, the depth of drains at different depths was analyzed and evaluated. In order to check the reflection coefficient ($\alpha$) in the range of 0.65 to 2.87, compared to the data measured in the region, the coefficient of 0.65 was reported as an optimal value. In order to confirm the

results, the statistical parameters of MAE, RMSE, and σ in the best-selected model showed values of 1.78, 2.02, and 0.02. Other findings were reported as follows, which are favorable results compared to other research.

- The numerical model of Bouwer & Van Schilfgarrd was announced as the best-selected model in this project.
- The distance between the drains in the superior model was chosen to be 51.26 m, which is 15 m more than the previously measured values in the region.
- The depth of placement of the drains was determined to be 130 cm, which is 70 cm less than the previously implemented values.
- By increasing the distance between the drains and reducing the digging depth, a 40% reduction in project implementation costs has been reported.
- By increasing the distance between the drains and reducing the digging depth, an increase in efficiency by 60% has been reported due to the presence of a wide stone bed.
- Due to the proximity of the impervious layer to the ground surface, the best response in the performance of computational drains is 5-day rainfall, which is a very favorable performance compared to the previous measurement values that showed 1-day rainfall.

**Author Contributions:** Conceptualization, B.M.; methodology, M.N.; software, B.M.; validation, M.M.N.; formal analysis, S.K. and B.M.; investigation, B.M.; resources B.M. All authors have read and agreed to the published version of the manuscript.

**Funding:** This research received no external funding.

**Institutional Review Board Statement:** Not applicable.

**Informed Consent Statement:** Informed consent was obtained from all subjects involved in the study.

**Data Availability Statement:** In this article, the dataset belongs to an area in the center of Iran called Dasht Sanjan lands in Central Province, which includes more than 500 hectares of land. A communication axis has divided these lands into 2 northern and southern parts. The northern part includes construction facilities and the southern part includes agricultural lands. The main goal of this research is the correct direction and transfer of surface and underground water during daily rainfall. Also, the second goal is to present a new plan to implement the distances between the drains and the depth of the underground drains. The results can be seen in the conclusion section of the article. Also, all documents and calculations and numerical tables with more than 2000 sheets of studies can be provided through the first author of the article.

**Conflicts of Interest:** The authors declare no conflict of interest.

## Abbreviations

The following symbols are used in this paper:

| | |
|---|---|
| MAE | mean absolute error |
| RMSE | root mean squared error |
| $\sigma$ | standard deviation |
| $R$ | correlation coefficient |
| $L$ | depth of drainage (cm) |
| $t$ | soil type in the study blocks of the region |
| $\alpha$ | reaction modulus |
| $h_0$ | standard depth of the drain (cm) |
| $r$ | radius of study boreholes (m) |
| $h$ | water depth in study boreholes (cm) |
| $d$ | agrology borehole (cm) |
| $b$ | study blocks in the region (number) |
| $s$ | area of study blocks (m$^2$) |
| $K$ | hydraulic conductivity $\left( \frac{m}{day} \right)$ |
| $\beta_1$ | the calculated distance of drainage in the area (m) |
| $\beta_2$ | the distance of the measured drains (m) |

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
