# Peer review of "The Sensitivity Analysis of the Drainage Unsteady Equations against the Depth of Drain Placement and Rainfall Time at the Shallow Water-Bearing Layers: A Case Study of Markazi Province, Iran"

_water, doi:10.3390/w14172693_

Round 1

Reviewer 1 Report

1- There are grammatical problems in some parts of the article. The submitted article needs to be reviewed by a translator. 2- In the abstract and summary presented in this article, it is necessary to present the contents in a more transparent manner. 3- The geographical coordinates of the subject area are mentioned in a part of the article. It is necessary to refer to an authoritative reference in the text of the article. 4- Even though the disadvantages of the reverse well method are mentioned in this article, why is this method used to calculate the hydraulic conductivity coefficient? 5- The interpretation of the results of the amberthermic diagram is mentioned in a part of the article in order to determine the type of climate of the study area. It is suggested to add this diagram to the article for better understanding. 6- Numerical values of the distance between drains in different numerical models are given in Table No. 5 and the numerical values of the MAE-σ parameter are given in Table No. 6. Can you provide auxiliary tables (separate from this article) that confirm the correctness of these numbers? 7- In a part of the article, the meteorological data of the studied area is mentioned. The annual rainfall is 305 mm per year. Is it correct to check the amount of rainfall between 1996-2016? It is believed that this number needs to be revised. Can you provide a source that confirms the meteorological data in the mentioned years? 8- The numerical results obtained from the mathematical models presented in this article should be compared with the measured values in the study area.

Author Response

Please find the attached file for the response to your comments.

Best Regards

Reviewer 2 Report

The manuscript is good structure, but some comments must be clarify:

1- Add the abbreviation in the manuscript.

2- intorduction should be improved by using following papers:

- Performance enhancement of a solar still using magnetic powder as an energy storage medium‐exergy and environmental analysis

- Techno-Economic Study of a New Hybrid Solar Desalination System for Producing Fresh Water in a Hot–Arid Climate

3- It is suggested to use of the bullet for conclusion.

Author Response

Dear Reviewer

We have read the cases mentioned by the honourable judges. We have written the appropriate answer below for each question and made these changes in our article. We hope the obtained results will be approved by you and the judges.

Best Regards

Mohsen Najarchi

  • Add the abbreviation in the manuscript.
  • Answer: Explanations of symbols and abbreviations in the text of the article were modified.
  • It is suggested to use the bullet for the
  • Answer: In the conclusion section, the results of the research were modified as individual cases using the bullet in the text of the article
  • The introduction should be improved by using the following papers.
  • Answer: The suggestions of the respected referee regarding the use of some sources in the text of the article were followed and new references were added to the article.

Round 2

Reviewer 1 Report

Accept in its current form

Reviewer 2 Report

Accepted.